# OpenReview forum: "Applying Refusal-Vector Ablation to Llama 3.1 70B Agents"
_NeurIPS.cc/2024/Workshop/SafeGenAi — SafeGenAi Poster_

### Official Review · Reviewer_mRU9 · 2024-10-09
**Reviews for paper titled "Applying Refusal-Vector Ablation to Llama 3.1 70B Agents"**

**Rating:** 6
**Confidence:** 4

**Review:**

Novelty: This novel approach involves the ablation of refusal vectors—directions in the model's residual stream responsible for refusal behavior—thereby enabling the model to bypass safety mechanisms and complete harmful tasks.
Reasons to Accept: The method is more computationally efficient than other techniques.
Reasons to Reject: Expanding the range of harmful tasks to include more sophisticated forms of manipulation or physical-world tasks . This paper focuses on the Llama 3.1 models, but it does not explore whether the findings generalize to other generative models such as GPT-4 or BERT-based systems

---

### Official Review · Reviewer_kQ3P · 2024-10-09
**showing safety training does not generalize to agents, but with confusing tone and message**

**Rating:** 5
**Confidence:** 3

**Review:**

This paper identifies that safety training in chat mode often does not carry over to queries within agentic scaffolding. It generalizes a technique called refusal vector ablation to the scenario of agentic use. Refusal vector ablation conceptualizes refusal behavior as a one-dimensional axis, which can be increased (potentially refusing benign requests) or decreased (performing even malicious requests).

This paper is attack focused, and it reveals important issues with safety training. Not only does chat training not generalize to agents, but the training itself is in a sense only one-dimensional and can be easily reversed. The experiments in the paper use refusal vector ablation to increase the strength of safety training in different llama models, but the paper itself already shows that this is a fragile and somewhat ineffective technique. Furthermore, the paper mentions that over 700 "abliterated" models are already publicly available.

The experiments in this paper appear to be comprehensive, with simulated environments for many common user tasks. The example of an agent deciding whether to send cryptocurrency is a particularly interesting one.

Overall, the paper seems to mentioned several very pervasive problems in this space, including having lots of abliterated models, the one-dimensionality of safety training, the ease of creating agents from open source models, etc. While the specific problem solved in the paper is smaller and more technical, I feel that these high-level issues are very important and could be included more in the discussion.

Pros
- comprehensive experiments and new benchmark setup
- generalization of refusal vector ablation and important step since agents are higher risk than chatbots
- identification of many serious issues with safety training
- strong introduction

Cons
- tone of paper does not indicate serious engagement with serious safety training problems
- unable to tell whether the paper is concerned about the proliferation of unrestricted models
- confusing message about relying on refusal vector ablation in experiments even though it is proven to be fragile
- title and abstract somewhat confusingly mention 70B even though experiments were also performed on 8B

Nits
- line 53: This is the first point residual stream is introduced and it does not have a citation, even though the paper says "they show that"

---

### Official Review · Reviewer_th3c · 2024-10-10
**Applying Refusal-Vector Ablation to Llama 3.1 70B Agents**

**Rating:** 4
**Confidence:** 5

**Review:**

Strengths:

- The relationship between safety fine-tuning and agentic behavior is an interesting direction. It is an important problem in the agent system.

Weaknesses:

- The presentation of the paper could be improved. Section 2 is difficult to follow and would benefit from more explanations and better organization.
The experiments are not comprehensive enough to fully support the conclusions in the paper. The experiments in Section 3 need to explore the impact of refusal-vector ablation on model safety more in-depth.
- Experiments should include a range of models. For instance, Llama-3-8B could help validate the generalizability of the conclusions across different model scales.